# Automated segmentation and quantitative analysis of organelle morphology, localization and content using CellProfiler

**Sebastiaan N. J. Laan** [1,2], **Richard J. Dirven** [1], **Petra E. Bürgisser** [2], **Jeroen Eikenboom** [1], **Ruben Bierings** [2]*, **for the SYMPHONY consortium** [¶]

1 Internal Medicine, Division of Thrombosis and Hemostasis, Leiden University Medical Center, Leiden, The Netherlands, 2 Hematology, Erasmus University Medical Center, Rotterdam, The Netherlands

¶ The complete membership of the author group (SYMPHONY consortium) can be found in the Acknowledgments.

* r.bierings@erasmusmc.nl

## Abstract

One of the most used and versatile methods to study number, dimensions, content and localization of secretory organelles is confocal microscopy analysis. However, considerable heterogeneity exists in the number, size and shape of secretory organelles that can be present in the cell. One thus needs to analyze large numbers of organelles for valid quantification. Properly evaluating these parameters requires an automated, unbiased method to process and quantitatively analyze microscopy data. Here, we describe two pipelines, run by CellProfiler software, called OrganelleProfiler and OrganelleContentProfiler. These pipelines were used on confocal images of endothelial colony forming cells (ECFCs), which contain unique secretory organelles called Weibel-Palade bodies (WPBs), and on early endosomes in ECFCs and human embryonic kidney 293T (HEK293T) cells. Results show that the pipelines can quantify the cell count, size, organelle count, organelle size, shape, relation to cells and nuclei, and distance to these objects in both endothelial and HEK293T cells. Additionally, the pipelines were used to measure the reduction in WPB size after disruption of the Golgi and to quantify the perinuclear clustering of WPBs after triggering of cAMP-mediated signaling pathways in ECFCs. Furthermore, the pipeline is able to quantify secondary signals located in or on the organelle or in the cytoplasm, such as the small WPB GTPase Rab27A. Cell profiler measurements were checked for validity using Fiji. To conclude, these pipelines provide a powerful, high-processing quantitative tool for the characterization of multiple cell and organelle types. These pipelines are freely available and easily editable for use on different cell types or organelles.

## Introduction

Eukaryotic cells are compartmentalized into organelles, subcellular entities separated from the cytoplasm by a limiting membrane that enable them to more efficiently carry out specialized functions in the cell, such as energy production and protein synthesis, transport and

**Data Availability Statement:** All images and raw data files are available from the Zenodo database (10.5281/zenodo.7875537).

**Funding:** This research received funding from the Netherlands Organization for Scientific Research (NWO) in the framework of the NWA-ORC Call grant agreement NWA.1160.18.038 https://symphonyconsortium.nl/ (SYMPHONY: personalized treatment for patients with a bleeding disorder) (received by SNJL, RB and JE) and from the Landsteiner Stichting voor Bloedtransfusie Research (LSBR-1707) https://lsbr.nl/ (received by RB). The funders had no role in study design, data collection and analysis, decision to publish, or preparation of the manuscript.

**Competing interests:** The authors have declared that no competing interests exist.

**Abbreviations:** A.U., Arbitrary intensity units; ECFC, Endothelial colony forming cell; HEK293T, Human embryonic kidney 293T; MTOC, Microtubule organizing center; OCP, OrganelleContentProfiler; OP, OrganelleProfiler; PDI, Protein disulfide isomerase; VWD, Von Willebrand disease; VWF, Von Willebrand factor; WPB, Weibel-Palade body.

degradation. A specific class of organelles consists of secretory vesicles, which serve to temporarily store and then rapidly secrete molecules into the extracellular space on demand. Secretory organelles are vital to maintaining homeostasis, as they allow a cell to communicate with other, distant cells or to respond to immediate changes in its environment, such as in the case of injury or when encountering pathogens. Their function is often defined by the content that is secreted, which is cell type and context specific, and depends on a sufficient magnitude of release, which directly relates to the number and dimensions of the secretory organelles that can undergo exocytosis. Moreover, the intracellular location of secretory organelles in relation to their site of biogenesis (i.e. the Golgi apparatus), filaments of the cytoskeleton and the plasma membrane also indirectly determines their exocytotic behavior.

Weibel-Palade bodies (WPBs) are cigar-shaped endothelial cell specific secretory organelles that contain a cocktail of vasoactive molecules that are released into the circulation in response to vascular injury or stress [1]. WPBs owe their typical elongated morphology to the condensation of its main cargo protein, the hemostatic protein Von Willebrand factor (VWF), into organized parallel tubules that unfurl into long platelet-adhesive strings upon release [2]. The size and shape of WPBs are of interest from a biological and medical perspective as they correlate with the hemostatic activity of the VWF strings that are released [3] and can be reflective of disease states, such as in the bleeding disorder Von Willebrand Disease (VWD) [4]. A model frequently used to study the pathophysiology of vascular diseases like VWD is the endothelial colony forming cell (ECFC). A major advantage of this model is that ECFCs can be derived from whole blood of patients, which allows analysis of patient endothelial cell function, WPB morphology and secretion *ex vivo*. However, substantial phenotypic heterogeneity can exist between ECFCs [5, 6], which stresses the need for robust quantitative analytical methods to evaluate their phenotype.

One of the most used and versatile methods to study number, dimensions, content and localization of secretory organelles is confocal microscopy analysis. However, as with all biological samples, considerable variability exists in the number, size and shapes of secretory organelles that can be present in the cell. One thus needs to analyze large numbers of organelles while ideally collecting this information in such a manner that it can be analyzed in a cell-by-cell manner. The crowded intracellular environment in combination with optical and immunostaining limitations presents an additional, technical challenge to separate individual organelles, which often precludes analysis on single organelle detail. Proper evaluation of these parameters requires an automated, unbiased method to process and quantitatively analyze microscopy data.

Here we describe 2 pipelines developed in CellProfiler [7], a free, easy to use image analysis software that uses separate module-based programming, for the identification, quantification and morphological analysis of secretory organelles within endothelial cells. The automated analysis pipeline OrganelleProfiler (OP) segments cells, organelles, nuclei and cell membranes from microscopy images, quantifies number, location, size and shape of organelles and extracts these data per cell and relative to the location of nucleus and perimeter of the cell. The function of the OrganelleProfiler pipeline is demonstrated by automated analysis of WPBs in 2 previously established phenotypic classes of healthy donor ECFCs [6], which identifies clear differences in number, length, eccentricity and intracellular localization of WPBs, and by morphometric analysis of early endosomes in ECFCs and HEK293T cells. Furthermore, the OrganelleProfiler was able to measure reduction in WPB size after Golgi ribbon disruption and to quantify perinuclear clustering of WPBs after stimulation with cAMP-mediated agonists. A second pipeline, called OrganelleContentProfiler (OCP), expands on the capabilities of the OrganelleProfiler by offering additional modules to measure the intensity of proteins of the secretory pathway both inside and outside the WPB, which we illustrate by analyzing the

presence of the WPB GTPase Rab27A [8–10] and the endoplasmic reticulum marker protein disulfide isomerase (PDI).

Our CellProfiler pipelines provide robust and unbiased quantitative analysis tools for WPB morphometrics and can, with minimal adaptation, also be used to obtain quantitative data for other organelles and/or other cellular systems.

## Materials & methods

### Endothelial colony forming cells and ethical approval

The study protocols for acquisition of ECFCs were approved by the Leiden University Medical Center and Erasmus MC ethics review boards. Informed consent was obtained from 4 subjects in accordance with the Declaration of Helsinki. Healthy participants were 18 years or older and had not been diagnosed with or known to have VWD or any other bleeding disorder. ECFCs used in this study have previously been classified as group 1 or group 3 [6].

### Cell culture, immunofluorescence and image acquisition

General cell culture of Endothelial Colony Forming Cells (ECFCs) and HEK293T cells was performed as described [5, 11]. ECFCs and HEK293T cells were grown on gelatin- or collagen-coated glass coverslips (9mm) and left confluent for 5 days before fixing with 70% methanol on ice for 10 minutes or with 4% paraformaldehyde for 15 minutes as described previously [10]. Disruption of Golgi ribbons was done by exposure of ECFCs to 2 μg/ml of nocodazole (Sigma, M1404) for 46 hours. For triggering of cAMP-mediated signaling to induce perinuclear clustering of WPBs, post-confluent monolayers of ECFCs were treated with 10 μM Forskolin (Merck, F3917) and 100 μM 3-isobutyl-1-methylxanthine (IBMX) (Merck, I5879) for 30 minutes as described previously [12]. Samples for OrganelleProfiler were stained with antibodies against VWF, EEA1, VE-cadherin, β-catenin, TGN46 and nuclei were stained with Hoechst or DAPI (S1 Table for supporting information on antibodies). Samples for OrganelleContentProfiler were stained with Hoechst and antibodies against VWF, VE-cadherin and either Rab27A or PDI. After staining with appropriate fluorescently labeled secondary antibodies, coverslips were mounted using ProLong® Diamond Antifade Mountant (Thermo Fisher Scientific). Visualization of the cells for the OrganelleProfiler example was done using the Imagexpress Micro Confocal System using the 63x objective without magnification or with the Leica Stellaris 5 Low Incidence Angle using the 63x oil immersion objective. The OrganelleContentProfiler samples were imaged using the Zeiss LSM900 Airyscan2 upright confocal microscope using the 63x oil immersion objective. For both the OrganelleProfiler and OrganelleContentProfiler images a Z-stack was made which was transformed to a maximum Z-projection.

### CellProfiler-based pipelines for cell organelle analysis and manual scoring with Fiji

CellProfiler (version 4.2.1 at time of publication) was used, which can be downloaded from the CellProfiler website (www.cellprofiler.org). For the initial development of the pipelines, confocal images from 33 ECFC clones from several healthy donors were used. These ECFCs have previously been classified into separate phenotypic groups based on cellular morphology [6]. The final pipeline was tested on 5 tile scans from one group 1 and one clone belonging to group 3. Images have to be of high enough resolution that individual organelles can be identified and do not blur together. Magnification, laser intensity, detector sensitivity and other acquisition parameters should be the same for each image set. Image format has to be similar

as well. We recommend uncompressed TIFF files. Pipelines developed are available in S1 and S2 Files and have been deposited in our laboratory GitHUB repository (https://github.com/Clotterdam). Adjusted pipelines optimized for the use on endosome quantification (in HEK293T cells—S3 File and in ECFCs–S4 File), WPB quantification after Golgi ribbon disruption (S5 File) and after forskolin stimulation (S6 File) are also made available. To compare the CellProfiler measurements and validate these, manual scoring of cell count, cell surface area, WPB count, WPB length, and VWF and Rab27A intensity inside and outside the WPBs was performed using Fiji version 2.3.0 [13]. Scoring was performed by using the built in scale and drawing regions of interest per cell and per WPB.

## Statistical analysis

Output data of the OrganelleProfiler pipeline was compared by Mann-Whitney U test if data was not normally distributed and unpaired T test with Welch's correction was performed on normally distributed data. Data of the OrganelleContentProfiler pipeline was compared with RM one way ANOVA with Geisser-Greenhouse correction. Data are presented as median with min/max boxplot. Results with p value $< 0.05$ were considered statistically significant. P values are indicated on the graphs in the figures. Data analyses were performed using GraphPad Prism 9.3.1 (GraphPad Software, San Diego, CA, USA).

## Results

### Development of OrganelleProfiler (OP)—Automated identification and quantification of nuclei, cells and secretory organelles

Described here are the modules used in the OrganelleProfiler pipeline for the identification and measurement of endothelial cells, their nuclei and WPBs. The most important parameters and how these can be adjusted for use on other tissues for each module are mentioned in S7 File. Full explanations of other variables are available from the help function within the CellProfiler software or from the user manual on the CellProfiler website. For the development of OrganelleProfiler we used confocal images from 33 ECFC clones from several healthy donors. These ECFCs have previously been classified into separate phenotypic groups based on cellular morphology and showed clear differences in expression of cell surface markers, proliferation and storage and secretion of VWF [6]. Representative images of 1 clone of group 1 (top) and 1 clone of group 3 (bottom) ECFCs used for this study are shown in Fig 1. The CellProfiler modules that together form the OrganelleProfiler pipeline can be divided into 6 steps (Fig 2), which are described below.

**Step I—Input of images.** Firstly, images of interest are imported into the software. In this example, 5 images from two groups of ECFCs were compared. Each image has 3 channels, 1 for the nuclei staining (Hoechst), one for cell membrane staining (VE-cadherin) and a third channel for organelle specific staining (VWF) (Fig 1). Channels are separated at this point so that each channel is processed separately in the following steps.

**Step II, III and IV—Identification of nuclei, cell membranes, cells and organelles.** Second, the nuclei staining signal is smoothed and a threshold is applied for the identification of the nuclei as objects. This object, together with the smoothed cell membrane staining signal is used in step III for the identification of the whole cell as secondary object. The nuclei are used as a starting point from which the object propagates outward in all directions until it encounters a secondary signal, in this case the smoothed cell membrane. A third object is generated using the cell object. This third object consists of only the cell membrane which is needed in the OrgannelleContentPipeline. In parallel to steps II and III, step IV uses the organelle

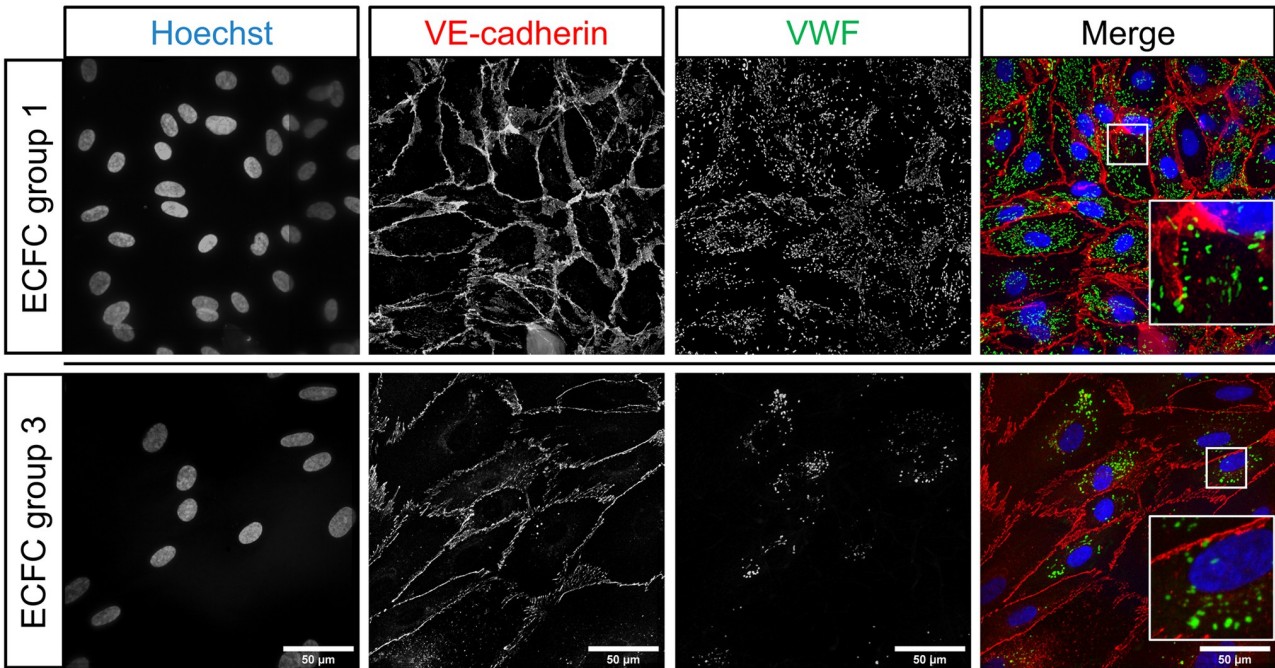

**Fig 1. Representative images of healthy ECFC controls belonging to previously classified groups based on morphology [6].** Group 1 ECFCs (top) and group 3 ECFCs (bottom) were stained with Hoechst (blue) and antibodies against VE-cadherin (red) and VWF (green). Scale bar represents 50 μm. Images were taken with a 63x objective.

staining signal for identification of the organelles. The signal is first rescaled and the speckle and neurite features are enhanced, which yields a better separation of organelles if they are located close to, or on top of, each other. After modification, the organelles are identified as the fourth object class.

**Step V—Measurement and relating of objects.** All objects that are generated in step II, III and IV are measured here. Size, shape and intensity, where relevant, is measured. Organelle objects are related to the nuclei and to the cell membrane in this step as well. This yields counts of secondary objects (organelles) per primary objects (cells) and distance of the secondary object to either the nuclei or the cell membrane. Measurements that we performed on the objects are eccentricity (as indicator for round or elongated WPB morphology), length of WPBs (maximum ferret diameter) and absolute as well as relative distance of WPBs to the nuclei and the cell membrane (Fig 3A).

**Step VI—Quality control and analysis of output.** For quality control, all objects' outlines are overlaid on the VWF signal. This overlay allows the user to check whether the pipeline was accurate in the identification of objects. Cells are numbered so potential outliers can be easily identified and the pipeline can be adjusted if needed. The exported output can be used to quantify and perform qualitative analysis on images of interest.

Automated quantification using OrganelleProfiler revealed significant differences in cell count, cell area and number, size, shape and localization of WPBs between group 1 and group 3 ECFCs (Fig 3B–3G). Fig 3B shows a significantly lower number of cells per image in group 3 (mean ± SD, 10.40 ± 2.40) compared to group 1 (37.60 ± 2.80) (p = 0.0003). Logically, as all ECFCs were confluent, we observed a larger mean cell area in group 3 (4016 ± 2445 μm$^2$) than in group 1 (1143 ± 516.60 μm$^2$) (Fig 3C) (p<0.0001). The total number of WPBs per image was lower in group 1 compared to group 3 (not shown). Additionally, the number of WPBs

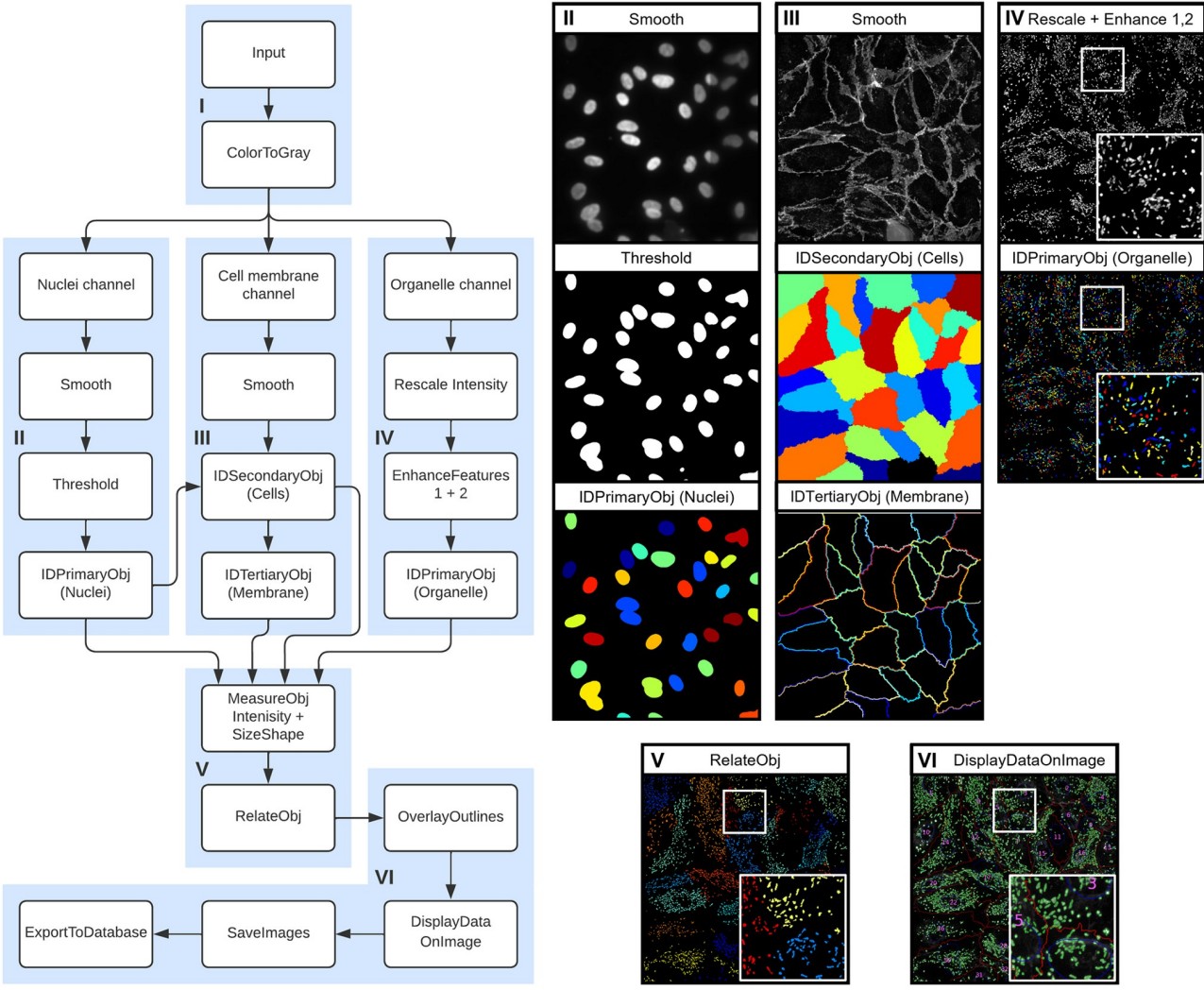

**Fig 2. OrganelleProfiler: Quantitative and qualitative analysis of cells and cell secretory organelles.** Left, flowchart of the modules within the OrganelleProfiler pipeline. I) Input of images and splitting of channels. II) Smoothing (top), thresholding (middle) and identification of the nuclei (bottom). Every different color indicates a different object. III) Smoothing of the cell membrane (top), identification of the cells (middle) and identification of cell membranes (bottom) as objects. IV) VWF signal rescaling and enhancement (top) and identification of WPB objects (bottom). V) Relating WPBs and Cells as child and parent respectively. Same colored objects indicate a relationship to the same cell. VI) Generated output image overlaying the outline of the nuclei (blue), cells (red), and WPBs (green) objects on the VWF channel. With the addition of the cell number (purple).

per cell was significantly lower in group 3 (30.92 ± 29.54) than in group 1 (107.30 ± 58.51) (p<0.0001) (Fig 3D). The distance of WPBs to the nuclei relative to their position in the cell was determined and shown in Fig 3E. The relative distance was significantly lower in group 3 ECFCs (32.31 ± 23.62%) when compared to group 1 (53 ± 30.10%) (p<0.0001) indicating that within the cell, WPBs were located closer to the nucleus in group 3 ECFCs. Finally, the mean WPB length was lower in group 3 (1.10 ± 0.27 μm) versus (1.38 ± 0.21 μm) in group 1 ECFCs (p<0.0001) and the WPBs were significantly more round in group 3 (0.63 ± 0.08)) versus (0.78 ± 0.04) (p<0.0001) (Fig 3F and 3G). The lower number of WPBs and the observation that they are smaller and rounder in group 3 when compared to group 1 could explain the decreased production and secretion of VWF observed previously [6].

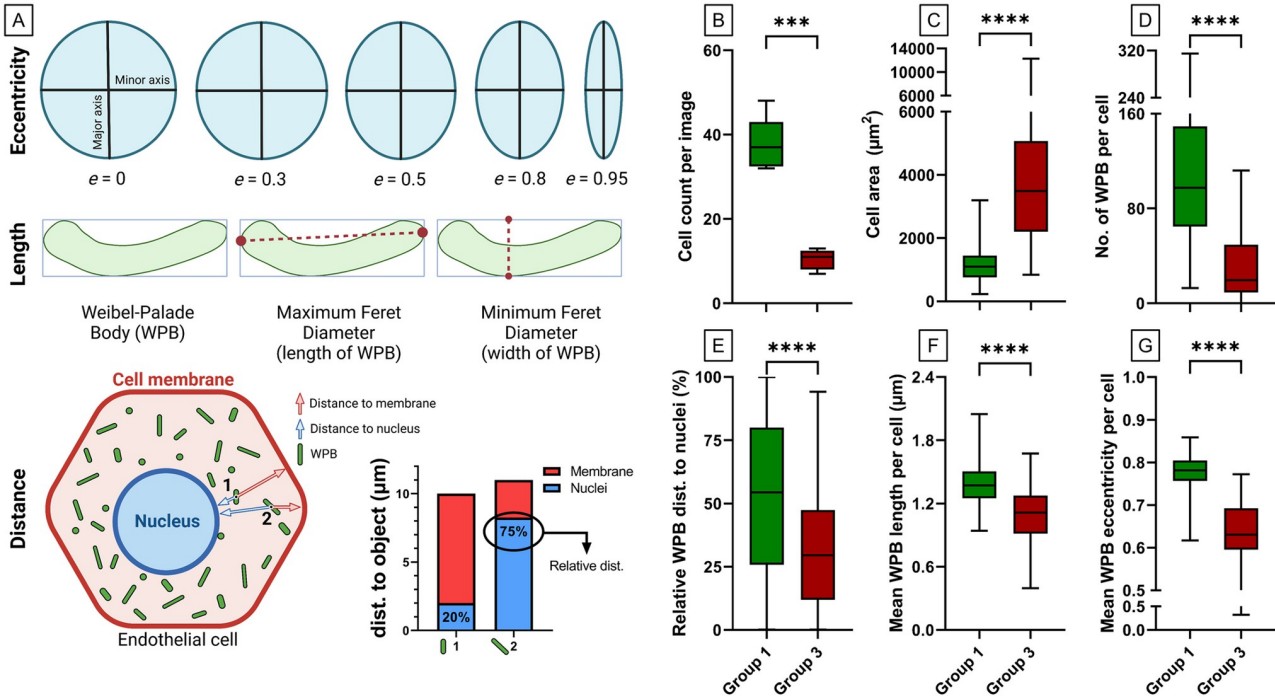

**Fig 3. Quantitative and morphological differences between ECFC control groups.** Two previously classified ECFCs based on morphology [6], group 1 (green) and group 3 (red), were stained for Hoechst, VE-cadherin and VWF. Per control, 5 images were analyzed with the OrganelleProfiler pipeline (each 44100 μm$^2$ in size). A) Graphical representation of the measurements that were performed on the objects. Eccentricity (top), length of Weibel-Palade bodies (WPBs) measured as maximum ferret diameter (middle) and distance of WPBs to the nuclei and the cell membrane was measured (bottom). Relative distance of the WPB to the nucleus in the cell was calculated as 100% x (distance to nucleus) / (distance to nucleus + distance to cell membrane). B) Cell count per image. C) The cell area (μm$^2$) per cell of all 5 images pooled (n = 188 in group 1 and n = 52 in group 3). D) Number of WPBs per cell. E) Distance of the WPB to the nucleus relative to their position in the cell in percentage. F) Mean WPB length per cell in μm. G) Mean eccentricity of WPBs per cell. Data is shown as median with min/max boxplot. Mann-Whitney U test was performed on not normally distributed data (D and G). Unpaired T test with Welch's correction was performed on normally distributed data (B, C, E and F); *p<0.05 **p<0.01, ***p<0.001.

To further validate the quantitative data obtained from our automated OrganelleProfiler pipeline we also performed a manual quantification of several of these parameters using Fiji image analysis software, specifically the region of interest manager [13]. One image of the group 1 ECFCs was used for the scoring. The manual scoring of the cells using the freehand selection resulted in 34 cells with a mean surface area of 1264 ± 497.93 μm$^2$. For three cells all WPBs were scored by measuring the longest distance in the WPB using a straight line. In these cells the manual scoring showed a mean WPB count of 117 ± 38.63 and a length of 1.57 ± 0.09 μm. All measurements were compared with the CellProfiler measurements on the same image and none of the results differed significantly. Taken together, we can conclude that both measurements with CellProfiler and Fiji are comparable and thus CellProfiler can be used to accurately measure cells and organelles.

## Validation and application of OrganelleProfiler

**Identification of HEK293T cells and early endosomes.** To demonstrate the versatility of the OrganelleProfiler pipeline, we extended our analyses to a different type of organelle (early endosome, visualized by staining for early endosome antigen 1, EEA1) and a different cell type (HEK293T). Representative images of ECFCs and HEK293T cells and the analyzed output of the OrganelleProfiler are shown in Fig 4A. The OrganelleProfiler was used to analyze three tile

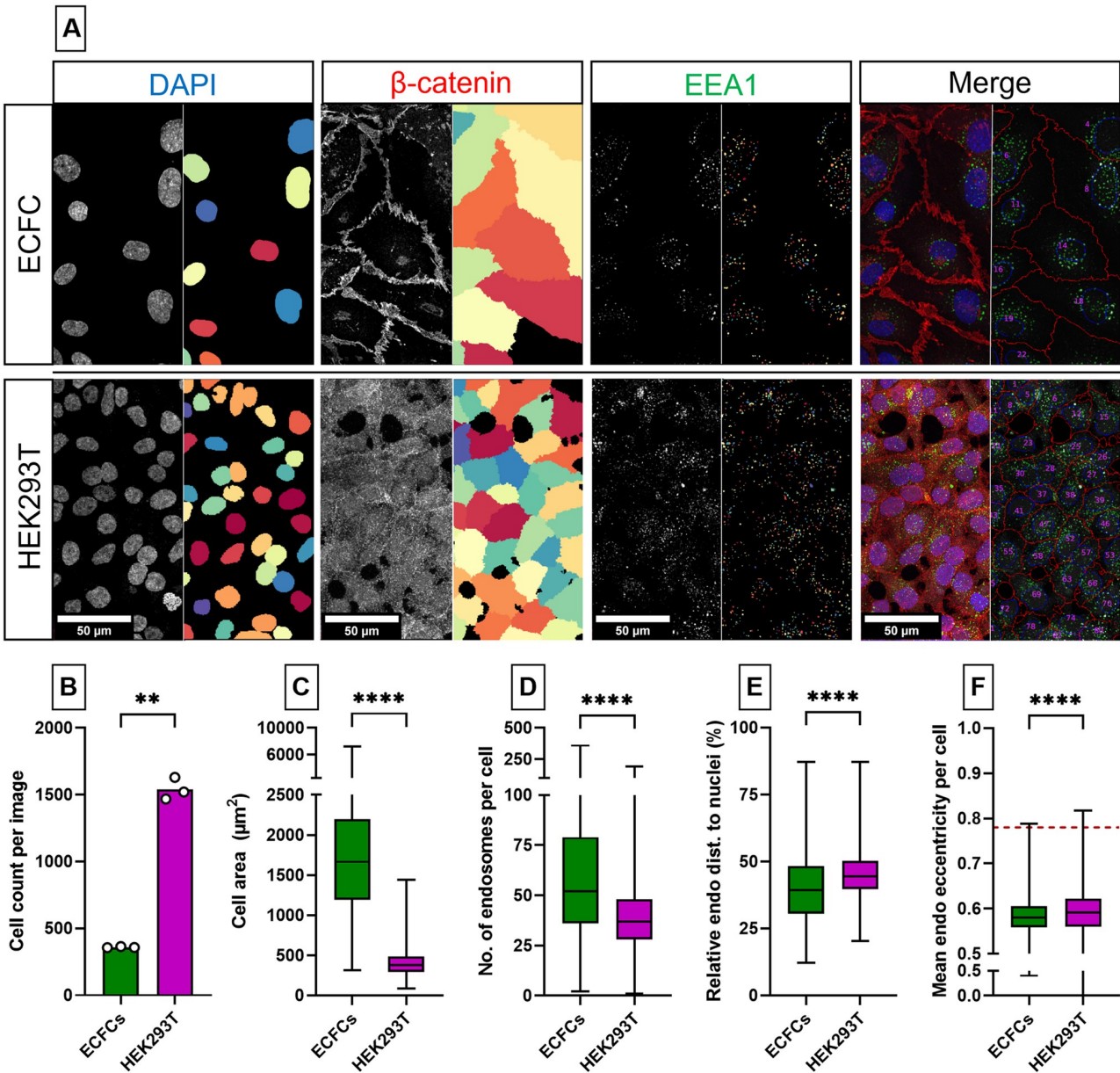

**Fig 4. Quantitative and morphological differences between early endosomes in ECFCs and HEK293T cells.** A) Representative images of ECFC (top) and HEK293T (bottom) cells where each panel shows the raw confocal image (left) and the identified object in Cellprofiler (right). Every different color indicates a different object. Cells were stained with DAPI (blue) and antibodies against β-catenin (red) and EEA1 (green). Scale bar represents 50 μm. Images were taken with a Leica Stellaris 5 LIA fitted with a 63x objective. Per cell type, 3 tile scans (each 754630 μm$^2$ in size) were analyzed with the OrganelleProfiler pipeline. B) Cell count per image. C) The cell area (μm$^2$) per cell of all 3 tile scans pooled (n = 1078 ECFCs and n = 4618 HEK293T cells). D) Number of endosomes per cell. E) Mean relative distance of the endosome to the nucleus per cell. F) Mean eccentricity of endosomes per cell. Data is shown as median with min/max boxplot. Dashed red line represents mean eccentricity of WPBs as determined in Fig 3G. Mann-Whitney U test was performed on not normally distributed data (C,D,E,F). Unpaired T test with Welch's correction was performed on normally distributed data (B); **p<0.01, ***p<0.001, ****p<0.0001.

scans of each condition. Fig 4B shows significantly more HEK293T cells per image (1539 ± 82.71) than ECFCs (359.30 ± 4.16) (p = 0.0016) which were also smaller (397.8 ± 150 μm$^2$) than ECFCs (1793 ± 866.3 μm$^2$) (Fig 4C) (p<0.0001). The number of early endosomes per cell was slightly lower in HEK293T cells (39.27 ± 16.22) compared to ECFCs

(63.30 ± 41.02) (p<0.0001) (Fig 4D). The relative distance of the early endosomes to the nuclei (Fig 4E) was higher in the HEK293T cells (45.34 ± 8.29%) when compared to ECFCs (40.05 ± 12.70%) (p<0.0001). Finally, the mean early endosome eccentricity is slightly higher in HEK293T cells (0.59 ± 0.05) versus (0.58 ± 0.04) in ECFCs (p<0.0001) (Fig 4F). Comparatively, the early endosomes are clearly rounder when compared to WPBs (0.78 ± 0.04, indicated by the red dotted line, data from Fig 3G) which are more elongated organelles.

**WPB shortening after Golgi ribbon disruption and perinuclear clustering after stimulation.** WPB length directly relates to the integrity of the Golgi ribbon: fragmented Golgi ribbons generate small WPBs while the longest WPBs require an extended, intact Golgi ribbon [3, 14–16]. Pharmacological inhibition of microtubules using nocodazole can be used to unlink Golgi ribbons [17] and this has been shown to reduce the size of newly generated WPBs [14, 18]. We used OrganelleProfiler to quantify the reduction of WPB length in ECFCs that were treated for 46 hours with vehicle (DMSO, control) or 2 µg/ml nocodazole. Staining for VWF and the trans-Golgi network marker TGN46 revealed a clear disruption of the Golgi in nocodazole-treated cells (Fig 5A). The OrganelleProfiler was used to analyze three tile scans of each condition. Nocodazole exposure caused a lower cell count per image (Fig 5B) (n = 184.7 ± 5.51 versus n = 553 ± 36.43, p = 0.0027) with larger surface area (2859 ± 1314 µm$^2$) than control ECFCs (1673 ± 641.6 µm$^2$) (Fig 5C) (p<0.0001). In the nocodazole treated cells we observed significantly rounder (nocodazole: 0.76 ± 0.06 vs control: 0.87 ± 0.02; p<0.0001) and shorter WPBs (nocodazole: 1.17 µm ± 0.18 vs. control: 1.54 ± 0.17 µm; p<0.0001) compared to the control cells (Fig 5D and 5E). The relative distance of the WPBs to the nuclei (Fig 5E) was slightly higher after nocodazole exposure (62.25 ± 13.77%) when compared to control (60.18 ± 8.17%) (p<0.0018).

It has previously been shown that, upon activation with cAMP-mediated agonists such as epinephrine or forskolin, endothelial cells cluster a subset of their WPBs at the microtubule organizing center (MTOC) via retrograde microtubular transport that depends on the minus-end motor protein dynein [19, 20]. We used OrganelleProfiler to quantify the reorganization of WPBs in forskolin-stimulated ECFCs (Fig 6A). Quantitative data gathered from one tile scan per condition shows that the per cell mean relative distance to the nucleus, which is directly adjacent to the MTOC and can therefore be used as a surrogate reference point for clustered WPBs, is lower in forskolin-treated cells (50.67 ± 14.76%, n = 352 cells) than in untreated cells (58.15 ± 10.22%, n = 537 cells) (Fig 6B, top). This indicates that in forskolin-treated cells, WPBs have on average moved towards the nucleus and thus into the direction of the MTOC. Also, when quantified on a single organelle basis, we see a shift towards lower relative distance to the nucleus in forskolin-treated cells (45.32 ± 31.39%, n = 17241 WPBs) compared with untreated cells (54.03 ± 30.77%, n = 21866 WPBs) (Fig 6B, bottom).

## OrganelleContentProfiler (OCP)—Automated measurement of proteins in secretory organelles

The OrganelleContentProfiler pipeline is an addition to the OrganelleProfiler pipeline. By adding 4 extra steps, secondary proteins of interest in, on or outside the organelle can be measured. For this purpose we analyzed the presence of Rab27A, a small GTPase that promotes WPB exocytosis and that is recruited to the WPB membrane during the maturation of these organelles after their separation from the Golgi complex [9, 10, 21]. We also determined, as a control, the presence of protein disulfide isomerase (PDI), a marker for the endoplasmic reticulum which should not show specific localization in or on the WPBs [5, 15]. Fig 7 shows example images of Rab27A as well as PDI co-staining in group 1 healthy donor ECFCs that were used in this pipeline. The CellProfiler modules that together form the

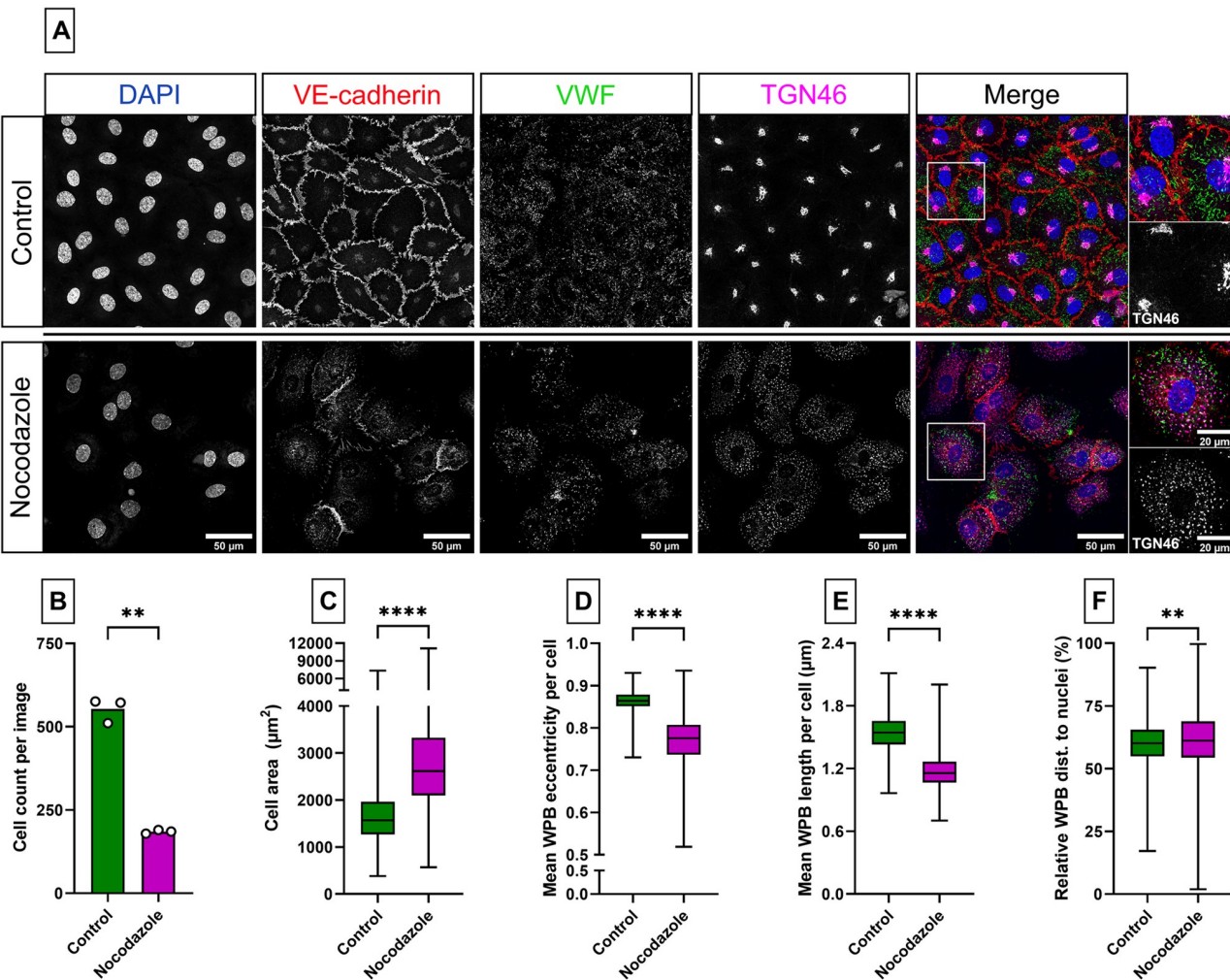

**Fig 5. Microtubule disruption leads to Golgi fragmentation and smaller and rounder WPBs.** A) Representative images of ECFCs treated for 46 hours with vehicle (control, top) or 2 μg/ml nocodazole (bottom). Cells were stained with DAPI (blue) and antibodies against VE-cadherin (red), VWF (green) and TGN46 (magenta). Scale bar represents 50 μm and 20 μm in the zoom in. Images were taken with a 63x objective. Per cell type, 3 tile scans (each 936054 μm² in size) were analyzed with the OrganelleProfiler pipeline. B) Cell count per image. C) The cell area (μm²) per cell of all 3 tile scans pooled (n = 1659 in the control and n = 554 in the nocodazole treated cells). D) Mean eccentricity of WPBs per cell. E) Mean length (μm) of WPBs per cell F) Mean relative distance of the WPBs to the nucleus per cell. Data is shown as median with min/max boxplot. Mann-Whitney U test was performed on not normally distributed data (C,D,E,F). Unpaired T test with Welch's correction was performed on normally distributed data (B); **p<0.01, ***p<0.001, ****p<0.0001.

OrganelleContentProfiler pipeline can be divided into 4 steps (Fig 8), which are described below. Further details on every module are described in S7 File.

**Step I—Input of an additional channel.** Similarly to the OrganelleProfiler, images are imported into the software. In this example, images have one additional channel containing the staining for either Rab27A or PDI. Again, channels are separated and the fourth channel is rescaled in order to view the channel in the final quality control.

**Step II—Import of organelle object identified in the OrganelleProfiler pipeline.** In this step, the organelle object as identified in the OrganelleProfiler pipeline is modified. The objects are initially identified using the staining for VWF, which is a cargo protein that is contained within the organelle. The secondary protein of interest, Rab27A, is a membrane protein that is

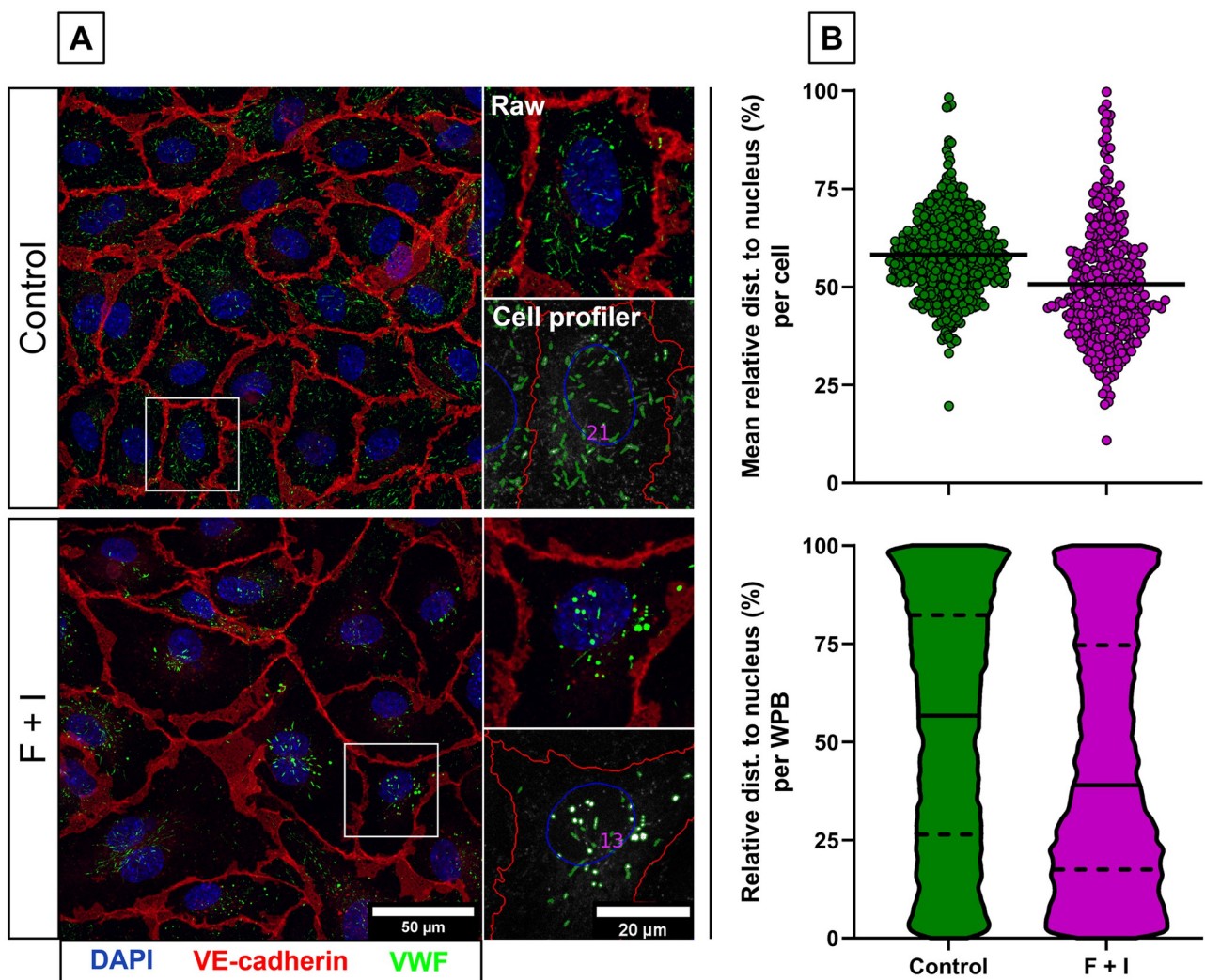

**Fig 6. Perinuclear clustering of WPBs after cAMP-mediated signaling in ECFCs.** A) Representative images of ECFCs treated for 30 minutes with vehicle (control, top) or 10 uM Forskolin and 100 uM IBMX (F+I, bottom). Cells were stained with DAPI (blue) and antibodies against VE-cadherin (red) and VWF (green). Scale bar represents 50 μm and 20 μm in the zoom in. Images were taken with a 63x objective. Per condition, 1 tile scan (970221 μm$^2$ in size) was analyzed with the OrganelleProfiler pipeline. B) Mean relative distance of the WPBs to the nucleus per cell (top) (n = 537 in the control and n = 352 in the stimulated ECFCs) and per WPB (bottom) shown as violin plot (n = 21866 for the control and n = 17241 for the stimulated ECFCs). The black bar indicates the median with quartiles.

located on the cytoplasmic face of the WPB membrane. To ensure full encapsulation of the Rab27A signal the object is therefore expanded by 2 pixels in all directions.

**Step III and IV—Identification of the secondary protein of interest "inside" and outside the organelle.** In these parallel steps, the expanded organelle objects and the rescaled secondary staining channel are used. The expanded objects are used as a mask to remove all signal of the Rab27A or PDI staining outside the organelle (step III) and inside the organelle (step IV). The remaining signal is then identified as object, resulting in two new objects containing the signal inside the organelles and outside the organelles respectively. These new objects are processed according to step V from the OrganelleProfiler including the measurements, relating, quality control and export.

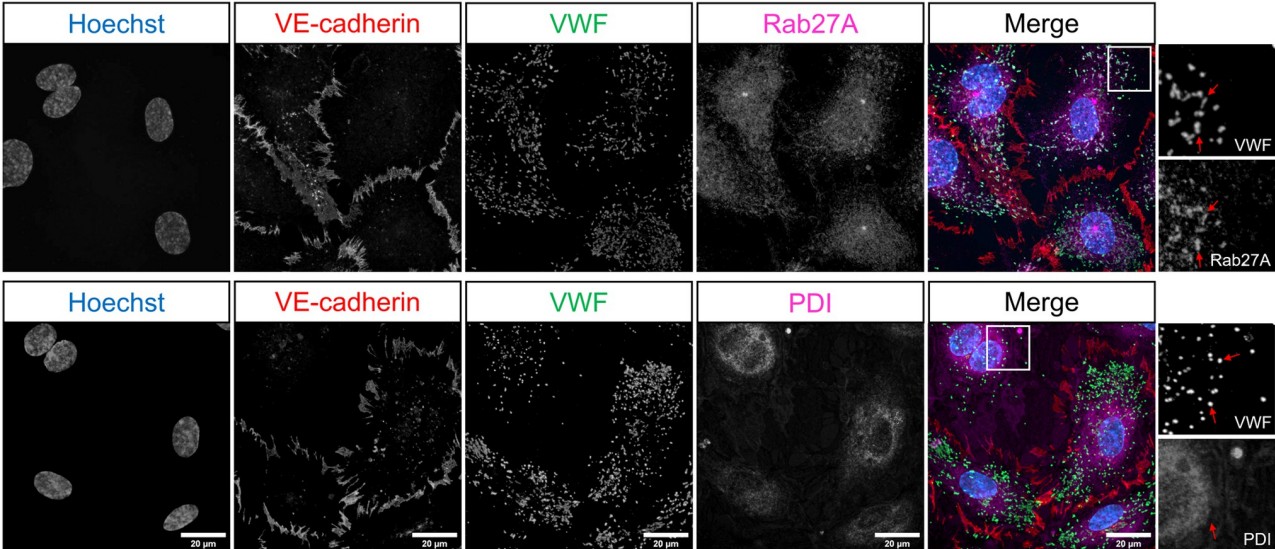

**Fig 7. Representative images of one healthy group 1 ECFC control belonging to previously classified groups based on morphology [6].** Cells were stained for Hoechst (blue), VE-cadherin (red), VWF (green) and Rab27A (top) or PDI (bottom). Scale bar represents 20 μm. Images were taken with a 63x objective. Red arrows indicate WPBs as identified in the VWF channel and the same location in the Rab27A or PDI channel.

**Step V and VI—Measurements, quality control and analysis of results.** In the OrganelleContentProfiler pipeline, different stainings on the same ECFC control are compared. In addition to the output from the OrganelleProfiler, the OrganelleContentProfiler provides measurements of the intensity of a secondary signal inside the organelle. Furthermore, it can quantify the cytoplasmic intensity values outside of the organelle which can be used for correction of the "inside" organelle signal. Signal intensity is noted as arbitrary intensity units (A.U.) as microscopes are not calibrated to an absolute scale.

We first confirmed that the number of WPBs quantified using OrganelleContentProfiler does not depend on the co-staining used (Rab27A: 204.8 ± 70.48; PDI: 146, ± 56.38; p = 0.24) (Fig 9A). ECFCs were stained with Hoechst and with antibodies against VE-cadherin, VWF and Rab27A or PDI. Fig 9B shows the A.U. inside and outside organelles and the A.U. inside the organelle corrected for the outside value. First, the VWF A.U. was analyzed as a measurement of a protein that is located predominantly in the WPB. The results show that the VWF A.U. values outside the WPBs was nearly zero (0.00076 ± 0.00033) and differed significantly from the inside A.U. (0.028 ± 0.0040) (p = 0.0016) indicating that VWF is almost exclusively present in WPBs. Secondly, it was determined that the Rab27A staining shows a significantly higher A.U. inside (0.081 ± 0.0085) the WPBs when compared to the outside measurement (0.052 ± 0.0041) (p = 0.0062). From this it can be concluded that part of the Rab27A protein is present in or on the WPB. Finally, the A.U. of the PDI staining was analyzed. PDI is only present inside the endoplasmic reticulum and should not yield increased A.U. inside the WPB. Indeed, the A.U. inside (0.074 ± 0.016) and outside (0.062 ± 0.0082) the organelle were similar (p = 0.20), indicating that PDI is not located specifically in or on WPBs.

Once more, to validate the quantitative data obtained by CellProfiler, we also performed a manual scoring using Fiji for the A.U. of the Rab27A and VWF staining inside and outside of all WPBs (n = 199) in one cell. We observed that the results determined manually using Fiji (A.U. VWF inside = 0.039, VWF outside = 0.000029; Rab27A inside = 0.094, Rab27A outside = 0.053) lie within the same range as those determined by CellProfiler. This shows that the

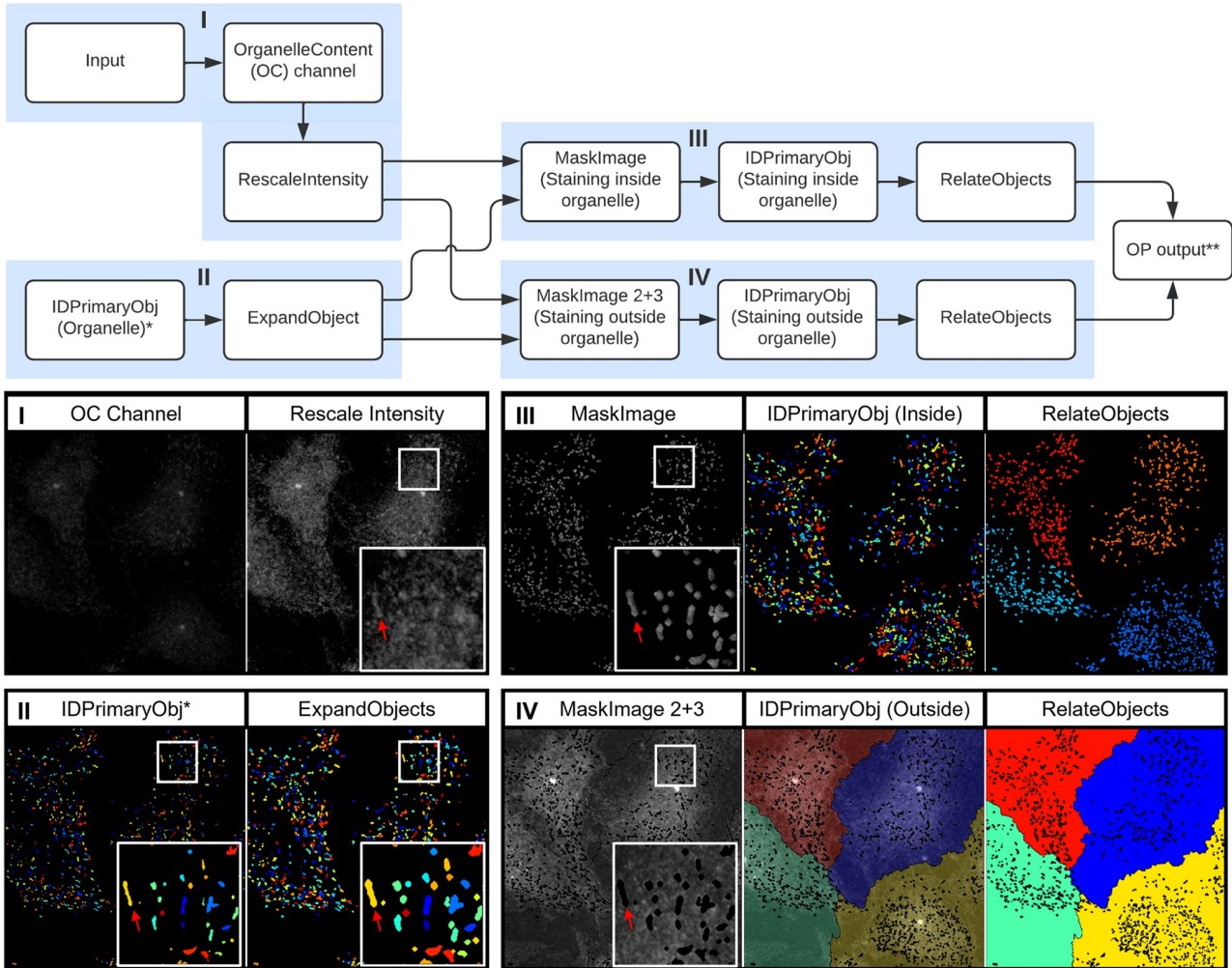

**Fig 8. OrganelleContentProfiler: Quantitative and qualitative analysis of other organelle proteins.** Top, flowchart of the modules within the OrganelleContentProfiler pipeline. I) Input of Rab27A (Organelle content) channel and rescaling of this channel. II) Input of primary object (Organelle) (left) and expansion of this object (right). III) Masking of the Rab27A channel using the Expanded organelle objects to leave only Rab27A signal inside the organelle (left). Identification of the Rab27A signal per WPB as object (middle) and relating these objects to the cells as child and parent respectively (right). IV) Masking of the Rab27A channel using the Expanded organelle objects to leave only Rab27A signal outside the organelle (left). Identification of the Rab27A signal in the cell as object without the WPBs (middle) and relating these objects to the cells as child and parent respectively (right). * Identified in step IV of the OrganelleProfiler pipeline (Fig 2). ** Pipeline continues with step V and VI from the OrganelleProfiler pipeline.

OrganelleContentProfiler can determine organelle specific stainings and measure the intensity of the staining corrected for the cytoplasmic value.

## Discussion

Quantifying large numbers of organelles is challenging due to the density and morphological heterogeneity of the organelles. The pipelines described here can be used to overcome these challenges and can provide organelle analysis in great detail on a larger scale. The Organelle-Profiler allows for measurement of cell and nucleus quantity and shape, and organelle quantity, shape, size and location within the cell. The organelles are also related to the cells which allows for cell-by-cell analysis. This information can be used to determine differences between

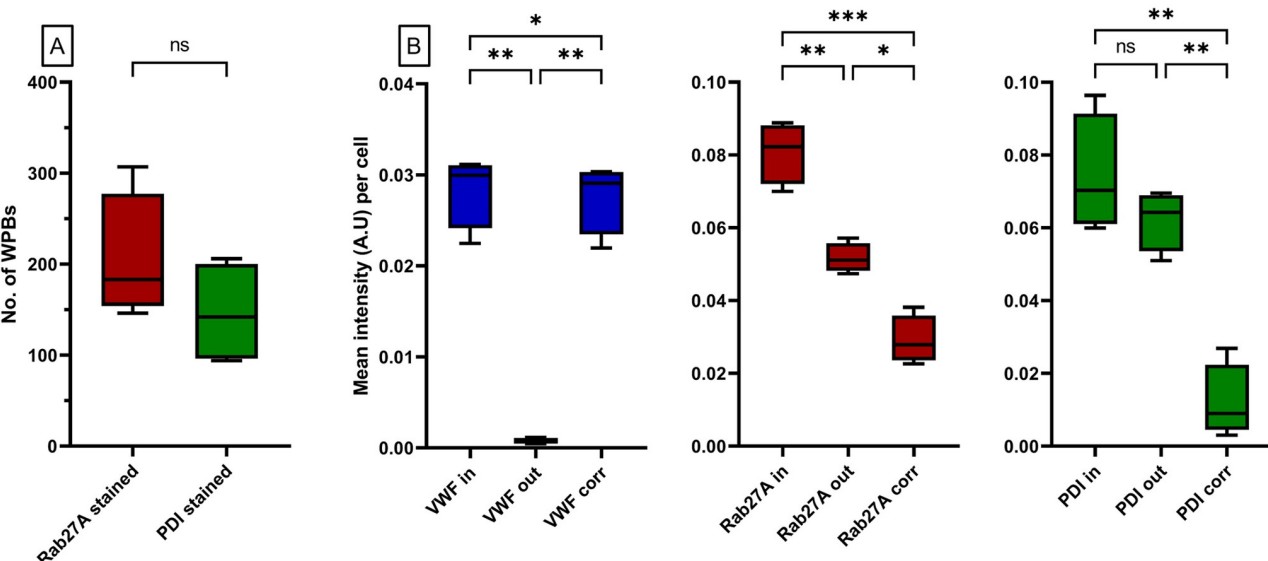

**Fig 9. Quantification of signal intensity inside cell organells.** A group 1 ECFC control as defined previously [6] was stained with Hoechst and with antibodies against VE-cadherin, VWF and PDI or Rab27A. One image (12769 μm²) per staining was analyzed using the OrganelleContentProfiler pipeline. A) Both images had the same number of cells (n = 4) and roughly the same number of WPBs. B) the mean intensity in arbitrary intensity units (A.U.) per cell for the PDI (left), Rab27A (middle) and VWF (right) staining. Each graph shows the measured mean intensity inside (in) the WPBs, outside (out) the WPBs and the intensity inside the WPB after correcting for the out signal (corr). Data is shown as median with min/max boxplot. RM one way ANOVA was performed with Geisser-Greenhouse correction; *p<0.05 **p<0.01, ***p<0.001.

a heterogeneous cell population or between patient and control cells. The OrganelleProfiler pipeline has shown significant differences between group 1 and group 3 ECFC controls based on only 5 images. Moreover, we were able to quantitatively determine differences in WPB length upon treatment with cytoskeletal drugs and differences in intracellular localization after induction of retrograde transport of WPBs. Once optimized for a set of images, the pipeline can analyze thousands of cells and hundreds of thousands of organelles within hours without potential bias associated with manual image processing and quantification. As shown with the early endosome staining, it is possible to adjust the pipeline to quantify other cell types and organelles. The pipeline was able to quantify differences between ECFCs and HEK293T cells and showed clear differences in eccentricity of the elongated WPBs compared to the round early endosomes.

With the OrganelleContentProfiler, secondary organelle markers can be measured and quantified. We showed 3 stainings of proteins with different localizations; PDI, which is only present on the endoplasmic reticulum, Rab27A which is present in the cytoplasm, but is also trafficked to the WPBs, and VWF which is mostly present in WPBs. Using the OrganelleContentProfiler pipeline we were able to quantify these stainings and determine the localization of these proteins. It is also possible to measure other organelle stainings at the same time by duplicating modules 3 to 8 of this pipeline and adjusting these for the additional channels.

Finetuning of the smoothing, thresholds and enhancement of the signal is necessary to ensure correct identification of objects. For every image set, a balance must be found to prevent over and under segmentation of organelles. Despite optimization, perfect segmentation of organelles is not always possible, especially in areas where organelles are crowded together. These imperfections may lead to incorrect identification of organelles, which could play out as underestimations of WPB numbers or overestimation of WPB dimensions. However, as all

images are analyzed by the same pipeline, this error is expected to occur to a similar extent in all samples. One point of improvement on the OrganelleContentProfiler pipeline could be the correction of the organelle secondary staining with the intensity levels directly surrounding the organelle instead of the mean intensity in the entire cytoplasm. This was not possible within the CellProfiler software but could be done in data processing afterwards using the MeasureObjectIntensityDistribution module and relating this to the distance of WPBs to the nucleus [7].

A comparison with manual scoring using Fiji was performed to check the validity of the results generated by our automated pipelines. We generally found that the results obtained with OrganelleProfiler and OrganelleContentProfiler correspond very well with manual quantifications using Fiji, although subtle differences were found for two parameters. First, the maximum ferret diameter is calculated based on the smallest convex hull that is created around the WPB. The manual scoring measured the length of the WPB in a line and not as the inside of a convex hull. This could cause the slight difference in length as measured between CellProfiler and Fiji. Second, VWF and Rab27A intensities inside WPBs as determined by manual scoring was slightly higher than from the OrganelleContentProfiler measurements. Possibly, the outlines that were drawn around WPBs manually were more strict than those generated by OrganelleContentProfiler, because the human eye is less capable at detecting the very small changes in signal intensity near the edges of the organelles. As such, the signal intensities in these edges may have not been included in the manual analysis, resulting in a higher mean value per WPB.

To conclude, the OrganelleProfiler and OrganelleContentProfiler pipelines provide powerful, high-processing quantitative tools for analysis of cell and organelle count, size, shape, location and content. These pipelines were created with the purpose of analyzing morphometric parameters of WPBs in endothelial cells, but as shown for the HEK293T cells and early endosomes, they can be easily adjusted for use on different cell types or organelles. This can be especially useful for analysis of large datasets where manual quantification of organelle parameters would be unfeasible.

## Supporting information

**S1 Table. Supporting information on antibodies.**
(DOCX)

**S1 File. OrganelleProfiler pipeline.**
(CPPIPE)

**S2 File. OrganelleContentProfiler pipeline.**
(CPPIPE)

**S3 File. OrganelleProfiler pipeline (adjusted and optimized for use on endosomes in HEK293T cells).**
(CPPIPE)

**S4 File. OrganelleProfiler pipeline (adjusted and optimized for use on endosomes in ECFCs).**
(CPPIPE)

**S5 File. OrganelleProfiler pipeline (adjusted and optimized for samples exposed to nocodazole).**
(CPPIPE)

**S6 File. OrganelleProfiler pipeline (adjusted and optimized for samples exposed to forskolin).**
(CPPIPE)

**S7 File. Detailed guide per module of the OP and OCP pipelines.**
(DOCX)

## Acknowledgments

The SYMPHONY consortium, which aims to orchestrate personalized treatment in patients with bleeding disorders, is a unique collaboration between patients, health care professionals, and translational and fundamental researchers specializing in inherited bleeding disorders, as well as experts from multiple disciplines [22]. It aims to identify best treatment choice for each individual based on bleeding phenotype. To achieve this goal, work packages (WP) have been organized according to 3 themes (e.g. Diagnostics [WPs 3 and 4], Treatment [WPs 5–9], and Fundamental Research [WPs 10–12]). Principal investigator: M.H. Cnossen; project manager: S.H. Reitsma.

According to Theme and workpackage: Prof. dr. M. de Haas, Dr. M. van den Biggelaar, Sanquin, Prof. dr. M.P.M de Maat, R.A. Arisz, Erasmus MC, Prof. dr. R.E.G. Schutgens, Dr. R. T. Urbanus, M. Zivkovic, UMCU, Prof. dr. F.W.G. Leebeek, Prof. Dr. H.F. Lingsma, E.S. van Hoorn, Erasmus MC, Prof. dr. R.A.A. Mathot, L.H. Bukkems, M.E. Cloesmeijer, A. Janssen, S. F. Koopman, Amsterdam UMC, M.C.H.J. Goedhart, L.G.R. Romano, W. Al Arashi, C. Mussert, Erasmus MC, Dr. S.C. Gouw, M.R. Brands, Amsterdam UMC, R.M.T. ten Ham, UMCU, D.M. Prameyllawati, Erasmus MC, Prof. dr. K. Meijer, UMCG, Prof. dr. A.L. Bredenoord, EUR, Dr. R. van der Graaf, L. Baas, UMCU, Prof. dr. J.J. Voorberg, Sanquin, Prof. dr. K. Fijnvandraat, Amsterdam UMC, Prof. dr. A.B. Meijer, J. Del Castillo Alferes, Dr. E. van den Akker, H. Zhang, Sanquin, Dr. R. Bierings, Erasmus MC, Prof dr. H.C.J. Eikenboom, LUMC, Dr. I. van Moort, Erasmus MC, S.N.J. Laan, LUMC.

Advisory Board: Prof. dr. C.A. Uyl-de Groot (EUR), C. Smit, G. Wijfjes (NVHP), Dr. M.H. A. Bos (NVTH), Prof. dr. V.W.V. Jaddoe (Erasmus MC), Prof. dr. P.J. Lenting (Université Paris-Saclay), Prof. dr. M. Makris (University of Sheffield), Prof. dr. Y.M.C. Henskens (MUMC).

## Author Contributions

**Conceptualization:** Sebastiaan N. J. Laan, Richard J. Dirven, Jeroen Eikenboom, Ruben Bierings.

**Data curation:** Sebastiaan N. J. Laan, Richard J. Dirven, Petra E. Bürgisser.

**Formal analysis:** Sebastiaan N. J. Laan.

**Funding acquisition:** Jeroen Eikenboom, Ruben Bierings.

**Investigation:** Sebastiaan N. J. Laan, Petra E. Bürgisser, Jeroen Eikenboom, Ruben Bierings.

**Methodology:** Sebastiaan N. J. Laan, Richard J. Dirven, Petra E. Bürgisser.

**Project administration:** Jeroen Eikenboom, Ruben Bierings.

**Resources:** Jeroen Eikenboom, Ruben Bierings.

**Software:** Sebastiaan N. J. Laan, Richard J. Dirven.

**Supervision:** Jeroen Eikenboom, Ruben Bierings.

**Validation:** Sebastiaan N. J. Laan.

**Visualization:** Sebastiaan N. J. Laan.

**Writing – original draft:** Sebastiaan N. J. Laan, Jeroen Eikenboom, Ruben Bierings.

**Writing – review & editing:** Sebastiaan N. J. Laan, Richard J. Dirven, Petra E. Bürgisser, Jeroen Eikenboom, Ruben Bierings.

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
