## [Decision Letter · Decision Letter 0]

2 Jan 2023

PONE-D-22-30713Automated segmentation and quantitative analysis of organelle morphology, localization and content using CellProfilerPLOS ONE

Dear Dr. Ruben Bierings,

Thank you for submitting your manuscript to PLOS ONE. After careful consideration, we feel that it has merit but does not fully meet PLOS ONE’s publication criteria as it currently stands. Therefore, we invite you to submit a revised version of the manuscript that addresses the points raised during the review process.

We have received one review of your paper from an external reviewer, and to speed up the process I have acted both as Academic Editor and as a Reviewer.

As you will see, the assessment of your study is in general positive but there are a number of issues that should be addressed before the paper is suitable for publication in PLOS ONE.

We look forward to receiving your revised manuscript.

Kind regards,

Julieta Alfonso, Ph.D.

Academic Editor

PLOS ONE

Journal Requirements:

Additional Editor Comments:

Review from Editor:

Laan et al. propose a method to automatically assess the number, morphology, and localization of intracellular organelles. To test the validity of the method, the authors used as a model endothelial forming cells that contain secretory organelles known as Weibel-Palade bodies.

The manuscript is well-written and contains valuable information for the cell biology community. However, the study focuses on one specific cellular model, which might limit the interest to a rather small scientific community. The authors claim that CellProfiler could be used for a variety of cell types and organelles. I encourage the authors to include yet another model for their analysis to validate the method in a wider range of cellular systems.

Reviewers' comments:

Reviewer's Responses to Questions

**Comments to the Author**

1. Is the manuscript technically sound, and do the data support the conclusions?

Reviewer #1: Yes

2. Has the statistical analysis been performed appropriately and rigorously? 

Reviewer #1: Yes

3. Have the authors made all data underlying the findings in their manuscript fully available?

Reviewer #1: Yes

4. Is the manuscript presented in an intelligible fashion and written in standard English?

Reviewer #1: Yes

5. Review Comments to the Author

Reviewer #1: In this manuscript Laan et al. have very elegantly described an automated method for the quantification of several parameter including length, distance from nucleus/ cell membrane, of subcellular organelles known as Weibel-Palade bodies (WPBs) in endothelial cells. This pipeline can be proven very useful in the production of unbiased data with morphometric charasterics of intracellular organelles including WPBs. Although the manuscript is very well written and sound, there are points in the manuscript that can be improved.

1. It is not very well explained how many ECFCs cell lines were used to develop this pipeline. Additionally, there is need for more clarification in how many images were used to create the pipeline and how many images the pipeline was tested onto. Could the authors provide this information?

2. Although Rab27a is a well known marker for mature WPBs, in the images shown there is also a lot of background that can skew their analysis. Do the authors have an explanation for that and also a different antibody to test. Another marker is CD63 that it have been shown to localize not only on lysosomes but also on WPBs. It would be very interesting to show this staining to as previous research has indicated CD63 to be not only localized in/ on WPBs but also in vesicles in WPBs. Can this pipeline distinguish these localizations?

3. Although the 2 groupd of ECFCs are clearly different in their morphology and content, which provides more evidence on the heterogeneity of ECFCs it would be interesting to show that the pipeline can distinguish the morphological parameters after treatment of eg group 1 ECFCs with nocodazole that disrupts the Golgi ribbon and results in shorter WPBs.

4. As the authors showed that the pipeline can very nicely indicate the distance of the organelle to the nucleus and cell membrane, it would be very interesting to show the distance to the nucleus and cell membrane after activation of ECFCs with different stimuli. The authors have previously shown very nicely that for example cAMP signaling can induce WPB perinuclear clustring. Does the pipeline can make these distinctions?

There are also 2 minor comments:

1. It is not shown on the graphs how many points there are per bar. When there are very big data sets even small differences can be statistically significant even though there is a minor change (eg length of the WPBs between group 1 and 3).

2. The authors showed there are less cells per field of view between gorup 1 and 3 with group 3 cells being significantly larger in cell area. Does correction for the larger cell area normalizes the cell count per field of view?

6. PLOS authors have the option to publish the peer review history of their article (what does this mean?). If published, this will include your full peer review and any attached files.

Reviewer #1: No

---

## [Author Response · Author response to Decision Letter 0]

28 Apr 2023

Dear sir/madam,

Below you can find our responses to each comment of the editor and reviewer. This information has also been added as the "Response to Reviewers" file.

Response to the editor and reviewer: 28-04-2023

We thank you for the feedback on our manuscript. The detailed comments were helpful and we believe they improved the clarity and impact of our study. Below we address the comments from the reviewers. We hope that the revised work meets the standards for publication. Our responses are noted in bold below the original comment/question. In the manuscript changes or additions are indicated in red.

Editor:

Laan et al. propose a method to automatically assess the number, morphology, and localization of intracellular organelles. To test the validity of the method, the authors used as a model endothelial forming cells that contain secretory organelles known as Weibel-Palade bodies.

The manuscript is well-written and contains valuable information for the cell biology community. However, the study focuses on one specific cellular model, which might limit the interest to a rather small scientific community. The authors claim that CellProfiler could be used for a variety of cell types and organelles. I encourage the authors to include yet another model for their analysis to validate the method in a wider range of cellular systems.

answer to editor:

We appreciate the positive comments and agree with the reviewer that the study's focus on one specific cellular model may limit its interest to a relatively small scientific community. Therefore, we have included a second cell model (HEK293T cells) and a different organelle (early endosomes stained with EEA1) in our analysis (figure 4). Our data shows that the OrganelleProfiler pipeline can accurately identify endosomes in both ECFCs and in HEK293T cells. Furthermore, with only minor adjustments the pipeline was optimized to accurately delineate individual HEK293T cells which, contrary to ECFCs do not form fully confluent monolayers. By doing so, we were able to validate the method in a wider range of cellular systems and demonstrate its versatility. Further details and data are noted in the manuscript. 

 

Reviewer #1: 

In this manuscript Laan et al. have very elegantly described an automated method for the quantification of several parameter including length, distance from nucleus/ cell membrane, of subcellular organelles known as Weibel-Palade bodies (WPBs) in endothelial cells. This pipeline can be proven very useful in the production of unbiased data with morphometric characteristics of intracellular organelles including WPBs. Although the manuscript is very well written and sound, there are points in the manuscript that can be improved.

answer to reviewer #1:

Thank you for your detailed review of our manuscript. We appreciate your comments and suggestions for improvement. Please find our responses to your questions below:

1. It is not very well explained how many ECFCs cell lines were used to develop this pipeline. Additionally, there is need for more clarification in how many images were used to create the pipeline and how many images the pipeline was tested onto. Could the authors provide this information?

answer 1:

We agree that information on the number of ECFC cell lines and images used to develop and test our pipeline was unclear. The pipeline was developed using a large group of 33 ECFC clones. The pipeline was then tested on 5 tile scans from 1 clone per group. This information is now adjusted/ added in the methods of the manuscript. 

2. Although Rab27a is a well-known marker for mature WPBs, in the images shown there is also a lot of background that can skew their analysis. Do the authors have an explanation for that and also a different antibody to test. Another marker is CD63 that it have been shown to localize not only on lysosomes but also on WPBs. It would be very interesting to show this staining to as previous research has indicated CD63 to be not only localized in/ on WPBs but also in vesicles in WPBs. Can this pipeline distinguish these localizations?

Answer 2:

The reviewer raises an interesting question regarding the Rab27A staining. For your first point, we have purposely chosen Rab27a because, depending on its activation state, it is present on WPBs (GTP-bound) or in the cytosol (GDP-bound) (Kat et al., Blood Advances, 2021). Since not the entire pool of Rab27A present in the cell is in its active state, the “background” Rab27A staining is actually to be expected. 

Regarding the second point, vesicles within the WPBs, such as the CD63-enriched intraluminal vesicles (ILVs) previously reported by Streetley et al. (Streetley et al., Blood, 2019) cannot be distinguished with the current pipeline. However, by adding modules, the pipeline could in principle be adjusted for this purpose by identifying the WPBs as parent object and using a secondary signal (e.g. CD63) to identify separate objects within the WPB. For this, it is important that the resolution should be sufficiently high to distinguish ILV-like structures within WPBs, which may require super resolution techniques such as Structured Illumination Microscopy. This is beyond the scope of this manuscript. However, our OrganelleProfiler and OrganelleContentProfiler are free to use and modify for specific research questions and we would like to encourage colleagues in the community to modify and expand the pipelines for their own requirements. 

3. Although the 2 groups of ECFCs are clearly different in their morphology and content, which provides more evidence on the heterogeneity of ECFCs it would be interesting to show that the pipeline can distinguish the morphological parameters after treatment of e.g. group 1 ECFCs with nocodazole that disrupts the Golgi ribbon and results in shorter WPBs.

Answer 3:

We have performed an additional experiment on group 1 ECFCs and stained WPBs after 46 hours of nocodazole exposure, which led to fragmented Golgis and smaller WPBs. The OP pipeline was able to successfully quantify the difference in WPB length and eccentricity. This data is added to the manuscript as figure 5.

4. As the authors showed that the pipeline can very nicely indicate the distance of the organelle to the nucleus and cell membrane, it would be very interesting to show the distance to the nucleus and cell membrane after activation of ECFCs with different stimuli. The authors have previously shown very nicely that for example cAMP signaling can induce WPB perinuclear clustering. Does the pipeline can make these distinctions?

Answer 4:

We have performed a stimulation assay on ECFCs using forskolin in combination with IBMX, which has previously been shown to induce WPB clustering in HUVECs (Rondaij et al, ATVB, 2006). After 30 minutes we observed clustering of the WPBs and using the OP pipeline we have quantified this difference between a control and stimulated condition. The data has been added to the manuscript and can be seen in figure 6.

There are also 2 minor comments:

1. It is not shown on the graphs how many points there are per bar. When there are very big data sets even small differences can be statistically significant even though there is a minor change (e.g. length of the WPBs between group 1 and 3).

Answer:

We apologize for not indicating the number of data points in our graphs. We have added this information in the figure legends to ensure transparency.

2. The authors showed there are less cells per field of view between group 1 and 3 with group 3 cells being significantly larger in cell area. Does correction for the larger cell area normalizes the cell count per field of view?

Answer:

The reviewer is right that there are less cells per field of view in group 3 than in group 1 (Figure 3B), although both formed confluent monolayers. Group 3 cells were on average significantly larger (Figure 3C) and indeed, when correcting for increased cell size the cell count normalizes.

---

## [Editor Report · Decision Letter 1]

3 May 2023

Automated segmentation and quantitative analysis of organelle morphology, localization and content using CellProfiler

PONE-D-22-30713R1

Dear Dr. Bierings,

Thank you for re-submission. You have successfully addressed all reviewer's concerns and I am very satisfied with the new version of your study.

We’re pleased to inform you that your manuscript has been judged scientifically suitable for publication and will be formally accepted for publication once it meets all outstanding technical requirements.

Kind regards,

Julieta Alfonso, Ph.D.

Academic Editor

PLOS ONE

---

## [Editor Report · Acceptance letter]

6 Jun 2023

PONE-D-22-30713R1 

Automated segmentation and quantitative analysis of organelle morphology, localization and content using CellProfiler 

Dear Dr. Bierings:

I'm pleased to inform you that your manuscript has been deemed suitable for publication in PLOS ONE. Congratulations! Your manuscript is now with our production department. 

Kind regards, 

on behalf of

Dr. Julieta Alfonso 

Academic Editor

PLOS ONE